# The VEGFA rs2010963 Gene Polymorphism Is a Potential Genetic Risk Factor for Myocardial Infarction in Slovenian Subjects with Type 2 Diabetes Mellitus

**DOI:** 10.3390/biom14121584

**Published:** 2024-12-11

**Authors:** Emin Grbić, Jernej Letonja, Danijel Petrovič

**Affiliations:** 1Department of Physiology, Faculty of Medicine, University of Tuzla, 75000 Tuzla, Bosnia and Herzegovina; emin.grbic@untz.ba; 2Laboratory for Histology and Genetics of Atherosclerosis and Microvascular Diseases, Institute of Histology and Embryology, Faculty of Medicine, University of Ljubljana, Korytkova 2, 1000 Ljubljana, Slovenia; jernej.letonja@mf.uni-lj.si; 3Institute of Histology and Embryology, Faculty of Medicine, University of Ljubljana, Vrazov trg 2, 1000 Ljubljana, Slovenia

**Keywords:** VEGFA, rs2010963, polymorphism, myocardial infarction, atherosclerosis, T2DM

## Abstract

Coronary artery disease (CAD) is a life-threatening condition caused by the chronic gradual narrowing of the lumen of the blood vessels of the heart by atherosclerotic plaque with a strong genetic component. The aim of our study was to investigate the association between the *VEGFA* polymorphism rs2010963 and myocardial infarction in patients with type 2 diabetes, as well as the expression of VEGFA. A total of 1589 unrelated Caucasians with T2DM lasting longer than 10 years were divided into two groups: case group subjects with MI (484) and a control group without a history of CAD (1105). A total of 25 endarterectomy sequesters were immunohistochemically stained to assess VEGFA expression. The rs2010963 polymorphism of the VEGFA gene was genotyped using a KBioscience Ltd. competitive allele-specific fluorescence-based PCR (KASPar) assay. The C allele was significantly more common in the case group according to the dominant model of inheritance (CC + CG vs. GG) (OR: 1.32; 95% CI: 1.05–1.66; *p* = 0.0197). A statistically significantly higher numerical areal density of VEGFA-positive cells was found in subjects with the C allele (CC + CG genotypes) in comparison to the GG genotype (117 ± 35/mm^2^ vs. 58 ± 21/mm^2^; *p* < 0.001). To conclude, the rs2010963 polymorphism is a potential genetic risk factor for myocardial infarction in Slovenian patients with T2DM.

## 1. Introduction

Coronary artery disease (CAD) is a life-threatening condition caused by the chronic gradual narrowing of the lumen of the blood vessels of the heart by atherosclerotic plaque, which leads to heart rhythm disorders, ischemic heart disease, and heart failure [1]. The pathogenic mechanisms of atherosclerosis include endothelial dysfunction and activation, adhesion, monocyte/macrophage activation and migration, local oxidative stress, lipid accumulation, extracellular matrix (ECM) synthesis, smooth muscle cell (SMC) migration and proliferation, and neovascularization within the plaque [2]. In regions affected by atherosclerosis, specific local conditions encourage the growth of new blood vessels from existing ones. This neovascularization enhances the local supply of nutrients and oxygen, potentially exacerbating the progression and remodeling of plaque. Furthermore, the incomplete maturation and fragility of these newly formed capillaries increase the risk of bleeding within the plaque, which can lead to plaque instability and rupture, and ultimately to acute myocardial infarction (AMI) [3]. However, the blockage of coronary blood vessels can also occur acutely with the bursting of a plaque and the formation of a blood clot, which will result in an AMI [4].

The prevalence of cardiovascular disease (CVD) is higher in patients who suffer from type 2 diabetes mellitus (T2DM) [5]. CAD and stroke are the most common causes of death in patients with T2DM [6]. The development of coronary atherosclerosis in people with type 2 diabetes (T2DM) is influenced by many modifying and non-modifying factors [7]. The process of atherosclerosis is accelerated in people with T2DM. Scientists around the world are increasingly searching for reliable markers with potential clinical effects [8].

A large number of genetic studies have proven the role of genetic factors in the development of CAD and AMI. Namely, numerous association studies link certain genes with preventive and risk factors for the occurrence of AMI [9]. Based on large genome-wide association studies (GWAS) studies, the heritability rate in CAD was found to be about 13.3%, and CAD was found to be associated with about 58 single-nucleotide polymorphisms (SNPs) [10].

Vascular Endothelial Growth Factor A (*VEGFA*) is a protein-encoding gene located on chromosome 6 on the p arm at position 21.1. This gene is part of the platelet-derived growth factor (PDGF)/VEGF growth factor family and encodes a heparin-binding protein, existing as a disulfide-linked homodimer [11]. This growth factor stimulates the growth and migration of vascular endothelial cells and is crucial for angiogenesis. The mutation of this gene in mice causes the improper formation of blood vessels at the embryonic stage [11]. Allelic variants are associated with complications of diabetes and atherosclerosis [12,13]. Different isoforms of this protein exist due to alternative transcript splicing and translation initiation. Animal experiments have also demonstrated a promising therapeutic use of VEGFA for cerebrovascular diseases; however, these promising effects have not been replicated in clinical trials [14]. In a study on the Chinese Han population, five SNP polymorphisms (rs699947, rs2010963, rs2071559, rs2305948, and rs1870377) of the VEGF and kinase insert domain receptor (KDR) genes were found to be associated with the occurrence of CAD [15]. In a meta-analytic study, an association was found between the polymorphism rs2010963 of the VEGF gene and CAD in the recessive (OR = 1.45, 95% CI = 1.03–2.05) and homozygous model (OR = 1.57, 95% CI = 1.02–2.42) [16].

The aim of this study was to investigate an association between the rs2010963 polymorphism of the *VEGFA* and myocardial infarction in patients with type 2 diabetes. Furthermore, we also explored VEGFA expression in coronary endarterectomy specimens with immunohistochemical staining.

## 2. Materials and Methods

### 2.1. Subjects

In the present case–control study, 1589 unrelated Caucasians with were included. An additional inclusion criterion for the participants in the control group was T2DM lasting longer than 10 years. Patients were divided into two groups: case group subjects with MI (484) and the control group without a history of CAD (1105). The control group of subjects consisted of subjects without ECG signs of ischemic disease and without ischemic changes during submaximal stress testing. Clinically silent CAD was not an exclusion criterion. T2DM was diagnosed according to the criteria of the American Diabetes Association [17]. The group of cases with a history of MI consisted of subjects who were included in the study 1 to 9 months after the acute event, based on established universal standards. In our study, only subjects with T2DM were included, due to the higher incidence of MI in this group compared to subjects without T2DM. The interview and physical examination were performed after informed consent was signed.

### 2.2. Ethical Statement

The study was approved by the National Medical Ethics Committee of Slovenia (number 0120-217/2019/8) and was structured in alignment with the principles outlined in the Declaration of Helsinki. All participants signed an informed consent form.

### 2.3. Biochemical Analyses

On an automated biochemical analyzer (Ektachem 250 Analyser, Eastman Kodak Company, Rochester, MN, USA), using standard colorimetric assays, we determined fasting glucose, total cholesterol (TC), low-density lipoprotein (LDL), high-density lipoprotein (HDL) cholesterol, and triglyceride (TG) values. The serum level of LDL was determined using the Friedewald formula. Total cholesterol values above 5 mmol/L or TG values above 2 mmol/L, or the use of hypolipidemic drugs, were defined as hyperlipidemia. To evaluate the value of glycosylated hemoglobin (HbA1c), we used high-performance liquid chromatography. The average value of HbA1c, obtained after three measurements, was used for the analysis. The level of high-sensitivity C-reactive protein (hsCRP) was measured using a Latex-enhanced immunonephelometric assay. Serum levels of cystatin c, urea, and creatinine were determined using standard biochemical analysis. The MDRD equation and creatinine values were used to estimate the glomerular filtration rate. Serum VEGF levels were determined using ELISA in a subpopulation of our participants (*n* = 100).

### 2.4. Genotyping

Genomic DNA was isolated from 200 μL of whole blood in the laboratory of the Institute of Histology and Embryology, Faculty of Medicine, University of Ljubljana. The isolation was performed using a QIAcube device (Qiagen GmbH, Hilden, Germany) and commercially available QIAamp DNA Blood Mini Kit (250) (Qiagen GmbH, Hilden, Germany) according to the “V3” protocol. The commercially available kit contained 96% ethanol, AW1 buffer, AW2 buffer, AE buffer, and AL buffer, as well as an appropriate amount of protease 285 µL/200 µL blood.

The rs2010963 polymorphism of the VEGFA gene was genotyped using KBioscience Ltd. (LGC, Teddington, UK) competitive allele-specific fluorescence-based PCR (KASPar) assay. The total volume of the reaction was 7 µL; the volume of DNA was 0.7 µL. The amplification protocol used was the “two-step touchdown” protocol with 36 cycles of amplification and an extra 10 cycles of recycling. Additional information is available at https://www.biosearchtech.com/support/faqs/kasp-genotyping-assays/ (accessed on 29 September 2024).

### 2.5. Immunohistochemistry

Endarterectomy sequesters were obtained from 25 subjects with T2DM with angina pectoris who had surgical myocardial revascularization. Coronary endarterectomy tissue samples were obtained during myocardial revascularization from diffusely diseased coronary arteries. Formalin-fixed paraffin-embedded tissue sections of endarterectomy sequesters were used for hematoxylin staining. Consecutive 5 μm tissue sections were cut from each paraffin block and then mounted and dried on glass slides. Tissues were deparaffinized and dehydrated in graded alcohol solutions. The slides were incubated with anti-VEGFA monoclonal antibodies (diluted 1:100, ThermoFisher, Waltham, MA, USA) overnight at 4 °C. The detection of VEGFA-positive cells was performed with the NovoLink Max Polymer Detection System (Leica Biosystems Newcastle Ltd., Newcastle upon Tyne, UK) following the manufacturer’s instructions. The cells were defined as VEGFA-positive/negative. The area with VEGFA-positive cells was manually marked and the numerical areal density of VEGFA-positive cells was calculated (the number of positive cells per mm^2^).

### 2.6. Statistical Analysis

For statistical analysis, we used SPSS version 26 (SPSS Inc., Chicago, IL, USA). Continuous variables with normal distribution were expressed as means ± standard deviation, while those with asymmetrical distribution were presented as median and interquartile range. The normality of continuous variables was assessed using the Kolmogorov–Smirnov test. Unpaired Student’s *t*-test was used for normally distributed continuous variables, whereas the Mann–Whitney U-test was used for asymmetrical ones. Discrete variables were compared using the Pearson χ^2^ test. Additionally, the Pearson χ^2^ test was applied to assess genotype distribution against Hardy–Weinberg equilibrium. Kruskal–Wallis test was used for assessing the relationship between the res2010963 genotypes and VEGFA serum levels. Fisher’s Exact test was used for establishing significant relationships between categorical variables in contingency tables with expected frequencies of lower than five. Stepwise multiple logistic regression was conducted for variables exhibiting a significant deviation in univariate analysis. A *p*-value less than 0.05 was considered statistically significant.

## 3. Results

Table 1 shows the basic anthropometric, biochemical, and clinical parameters of the case group (with MI) and the control group with T2DM. There were no significant differences between the groups with respect to body mass index (BMI), systolic blood pressure (SBP), fasting glucose, triglycerides (TG), glycated hemoglobin (HbA1c), concomitant history of transient ischemic attacks (TIA), and incidence of drugs affecting the renin–angiotensin–aldosterone system. The participants in the case group were older and more likely to be male, had a higher total cholesterol, had a higher hsCRP, had a higher percentage of former and active smokers, were more physically active, had a higher percentage of alcohol use, had a higher incidence of cerebrovascular stroke, and had a higher percentage of stenting and coronary artery bypass grafting (CABG). Moreover, the cases had a higher incidence of statin, aspirin, beta blockers, and hypoglycemia drug use than controls. Compared to the control group, the case group had lower diastolic pressure values, a shorter duration of T2DM, and a lower level of HDL–cholesterol. Participants in the case group had a lower estimated glomerular filtration rate (eGFR) and higher levels of creatinine, urea, and cystatin c.

The genotype and allele frequencies of the rs2010963 polymorphism of the VEGFA gene are shown in Table 2. Cases (with MI) and controls (without CAD) were in Hardy–Weinberg equilibrium regarding the genotype distribution (*p* = 0.14; *p* = 0.13, Pearson χ^2^ test, respectively). In the case group, the CC genotype was more frequent, but there was no statistically significant difference. However, the C allele was significantly more frequent in the case group.

We used logistic regression analysis to assess whether the rs2010963 polymorphism was independently associated with MI after adjusting for age, waist circumference, diastolic blood pressure, fasting glucose, total cholesterol, gender, smoking, waist circumference, urea, creatinine, cystatin c, and eGFR. The results in the dominant genetic model (*p* = 0.0197) indicate the existence of a statistically significant association (Table 3). The association was also statistically significant in the comparison between the CG and GG genotypes according to the co-dominant genetic model (*p* = 0.0404). The statistical strength of the study was 0.80.

There was no statistically significant difference in serum VEGFA levels between the participants in the case and control groups (Table 4), nor did we find an association between the rs2010963 genotypes and VEGFA serum levels (Table 5).

At the end of this study, the coronary artery segments, which were obtained via endarterectomy from subjects with advanced atherosclerosis, were examined using immunohistochemical staining. A statistically significantly higher numerical areal density of VEGFA-positive cells was found in 14 subjects with the C allele (CC + CG genotypes) (Figure 1) in comparison with 11 subjects with the VEGFA GG genotype (wild type) (117 ± 35/mm^2^ vs. 58 ± 21/mm^2^; *p* < 0.001).

## 4. Discussion

In our study, we investigated the association of VEGFA gene polymorphism rs2010963 with AMI in the Slovene Caucasians with T2DM. We found an increased risk in carriers of the C allele of the rs2010963 polymorphism for AMI in the dominant genetic model compared to T2DM patients with no known CAD (OR: 1.32; 95% CI: 1.05–1.66; *p* = 0.0197). An increased risk was also discovered in the participants with the CG genotype in comparison to the GG genotype in the co-dominant genetic model (OR: 1.29; 95% CI: 1.01–1.64; *p* = 0.0404). We also discovered that the numerical areal density of VEGFA-positive cells was greater in the participants with the CC and CG genotypes in comparison to the participants with the GG genotype. The results of our investigation warrant a comparative analysis with analogous or cognate studies, with an emphasis on exploring their potential implications for clinical application.

The lipid status of the participants in both groups was quite poor and evidently not optimally managed. There was a statistically significant difference between the LDL-c, HDL-c and total cholesterol levels between the groups. A larger percentage of participants in the case group were prescribed statins than in the control group (80.0% vs. 67.7%) and a higher proportion of participants in the case group reported regular physical activity (3–4 times/week) than in the control group. Poor adherence to therapy could partially explain the increased LDL-c levels in both groups.

Many studies worldwide, on different populations, have investigated the association of the rs2010963 polymorphism of the VEGFA gene with the onset of CHD, CAD, and AMI. Kutya et al. determined the predictive values of VEGFA polymorphism rs2010963, obtaining similar results to ours [18]. They divided the patients into two groups according to their genotype (GG vs. combined GC + CC). The results showed that the GC or CC genotypes of the rs2010963 polymorphism in the VEGFA gene are an independent predictor of clinical outcome. Also, they found that STEMI patients with the GC or CC *VEGFA* genotype had a higher incidence of the clinical combined endpoint of accumulation compared to those with the GG *VEGFA* genotype (*p* = 0.02) [18]. In the study conducted by Kobets et al. on hypertensive patients with STEMI, the role of the VEGFA gene polymorphism (rs2010963) was investigated. Their study was a one-year prospective study that included 91 patients. The plasma levels of VEGFA were found to be significantly higher at the beginning of the study in the participants with the GG genotype, but the levels significantly increased in the participants with the GC genotype after 1 year (*p* = 0.001) [19]. A study similar to the previous one confirms the significantly higher levels of VEGFA in participants with the GG genotype who survived a STEMI after one year of follow-up [20]. Compared to our research, this research had a prospective character and a smaller number of patients were included in the study. Also, the subjects of these studies were younger than ours.

Kalayi Nia et al. performed a study on the Iranian population, which included 520 subjects that were divided into two groups: 347 participants in the CAD group and 173 in the no-CAD (control) group. A 5-year follow-up of 484 subjects for cardiovascular-related outcomes, and the association of the rs2010963 polymorphism of the VEGF gene with the presence of CAD and long-term survival, were investigated. A significant association was found between the CC genotype of the rs2010963 polymorphism of the VEGFA gene and CAD (OR = 3.65, 95%CI = 1.53–8.72; *p* = 0.003). Also, the CC/CG rs2010963 genotypes showed an increased probability of developing CAD (*p* = 0.027). Furthermore, carriers of the C allele showed a higher probability of developing CAD compared to carriers of the G allele (*p* = 0.002) [21]. In relation to our research, this study had a similar genotype distribution and allele frequency in the CAD group as in our study, with a slight difference in the percentage of the CG genotype (Iranian study 38.9% vs. our study 47.9%), also the respondents in the Iranian study were relatively younger compared to our study. Further, many other studies linked the rs2010963 polymorphism of the VEGFA gene with an increased risk of developing MI or CAD in different populations [16,22,23,24,25].

Kangas-Kontio et al. investigated the association between three SNPs in the VEGF gene and AMI and carotid intima-media thickness. In their study, they included 516 participants of the OPERA (the Oulu Project Elucidating Risk of Atherosclerosis) study and 251 AMI survivors. They did not find an association between rs2010963, rs699947, or rs3025039 and AMI. The differences between the participants included in their study and those included in ours could explain the conflicting conclusions. Their participants were younger (only participants younger than 65 were included), were not all diabetics, and were of Finnish origin [26].

There was no statistically significant difference in serum VEGFA levels between the participants who suffered an MI and controls in the subpopulation of participants where an ELISA was performed. Several studies reported increased VEGF levels after MI in the first days/weeks after the event [27,28]. Blood samples from our participants were obtained between 1 and 9 months after the event and we hypothesize that the increased VEGFA levels normalized before the participants were included in our study and had their blood sample drawn.

An additional analysis of the association between rs2010963 SNP and serum VEGFA levels showed no association between the genotype and VEGFA levels. In the study by Palmer et al., no association was found of the rs2010963 polymorphism of the VEGFA gene with the level of VEGFA in plasma [29]. Similarly, Skrypnik et al. did not report an association between the VEGFA levels and the rs2010963 polymorphism [30]. It is known from most of the research that VEGFA levels are associated with the risk of CHD, CAD, or AMI; however, from our own results and these two studies, we can conclude that the association of the mentioned polymorphism does not correlate with the VEGFA levels. It is important to note that we only analyzed the association between the rs2010963 genotypes and VEGFA levels on a subpopulation of 100 participants.

It is well established that conditions such as T2DM and rheumatoid arthritis (RA) significantly increase the risk for CVD. The CC genotype of the rs2010963 polymorphism was associated with increased BMI and fasting blood glucose in the study conducted by Abbasalizad Farhangi et al. on an Iranian population [31]. Sellami et al. reported an association between the rs2010963 polymorphism and T2DM in a Tunisian Arab population. They included 815 patients with T2DM and 805 controls and also reported an association between the GC genotype of the rs2010963 SNP and decreased VEGF levels [32]. Several researchers have also reported an association between the rs2010963 SNP and CVD in patients with rheumatoid arthritis (RA) [33,34].

Our study also had several limitations. First of all, the sample was very homogeneous and all subjects were Caucasians. We only investigated one polymorphism in the VEGFA gene and did not account for the influence of other polymorphisms that could have affected our results. The patients in both groups used drugs for a longer period of time, so their effect on the result cannot be completely ruled out. Future studies that take into account the medication used by the participants, as well as the effect of other polymorphisms, are needed. Another important limitation of our study is the fact that we did not gather and analyze the urine of the participants. Knowing the albuminuria levels and albumin/creatinine ratios in the urine would allow for an even more accurate assessment of the kidney function of the participants. Also, we only determined the serum level of VEGFA on a subpopulation of 100 participants and we only performed immunohistochemical staining on a small sample (but we did not have more material available). Future studies should investigate the levels of VEGFA and the association between the VEGFA levels and rs2010963 polymorphism, and should include a larger sample size of atherosclerotic plaques to better assess the relationship between the rs2010963 polymorphism and VEGFA expression.

## 5. Conclusions

In our cross-sectional case–control study, we showed that the rs2010963 VEGFA gene polymorphism is associated with AMI in a Slovenian cohort with type 2 diabetes mellitus. In order to confirm the influence of the rs2010963 polymorphism on the occurrence of AMI, studies on a larger sample are needed, as well as more samples for immunohistochemical analyses of the VEFGA expression in heart blood vessels in different populations. It would also be important to investigate the VEGFA protein levels in the plasma of patients with and without CAD.

## Figures and Tables

**Figure 1 biomolecules-14-01584-f001:**
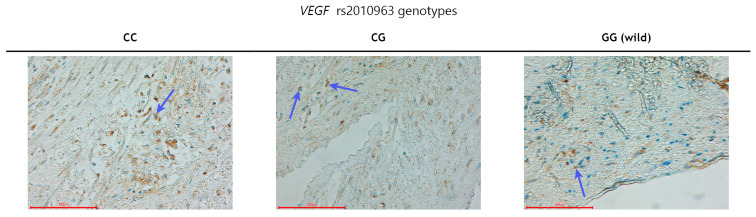
Immunohistochemical analysis of VEGFA protein expression relative to the rs2010963 genotypes. VEGFA-positive cells are stained brown; VEGFA-negative cells only have their nuclei stained blue. The blue arrows mark VEGFA-positive cells.

**Table 1 biomolecules-14-01584-t001:** Anthropometric, biochemical, and clinical characteristics of the case and control groups.

	Case (*n* = 484)	Control (*n* = 1105)	*p* Value
Age (years)	66.19 ± 9.78	64.10 ± 9.12	**0.001**
BMI (kg/m^2^)	29.57 ± 4.12	29.88 ± 4.84	0.22
Waist circumference (cm)	102.93 ± 12.17	108.05 ± 12.70	**<0.001**
SBP (mm Hg)	147.67 ± 19.87	148.39 ± 20.63	0.51
DBP (mm Hg)	81.74 ± 10.64	84.31 ± 10.42	**<0.001**
Fasting glucose (mmol/L)	8.79 ± 2.96	8.55 ± 2.60	0.10
Total cholesterol (mmol/L)	4.90 (4.00–5.90)	4.70 (4.00–5.60)	**0.029**
HDL-c (mmol/L)	1.10 (0.90–1.30)	1.20 (1.00–1.40)	**<0.001**
LDL-c (mmol/L)	2.70 (2.20–3.60)	2.70 (2.10–3.40)	**0.026**
TG (mmol/L)	1.80 (1.20–2.60)	1.70 (1.20–2.50)	0.32
HbA1c (%)	7.91 ± 1.33	7.77 ± 1.36	0.067
hsCRP (mg/L)	2.20 (1.20–3.60)	1.80 (1.10–2.90)	**<0.001**
T2DM duration (years)	11.00 (6.00–20.00)	15.00 (10.00–20.00)	**<0.001**
Gender	303 (62.6%)	557 (50.4%)	**<0.001**
Smoking (%)			**<0.001**
Never	386 (79.8%)	990 (89.6%)	
Former smoker	23 (4.8%)	10 (0.9%)	
Active smoker	75 (15.5%)	105 (9.5%)	
Phisycal activity (3–4-times per week)	366 (75.6%)	740 (67.0%)	**<0.001**
Alcohol	85 (17.6%)	102 (9.2%)	**<0.001**
CVI	58 (12.0%)	77 (7.0%)	**<0.001**
TIA	22 (4.5%)	31 (2.8%)	0.075
CABG	122 (25.2%)	0 (0.0%)	**<0.001**
Stenting of coronary arteries	163 (33.7%)	0 (0.0%)	**<0.001**
Statin use	387 (80.0%)	748 (67.7%)	**<0.001**
Hypoglycemic drugs	396 (81.8%)	756 (68.4%)	**<0.001**
Aspirin	193 (39.9%)	312 (28.2%)	**<0.001**
Beta blokers	194 (40.1%)	182 (16.5%)	**<0.001**
eGFR (mL/min/1.73 m^2^)	69.00 (7.75–88.25)	79.50 (60.00–90.00)	**0.0282**
Creatinine (mmol/L)	88.00 (73.00–106.00)	77.00 (64.00–93.00)	**<0.001**
Urea (mmol/L)	6.90 (5.50–8.70)	6.00 (4.90–7.60)	**<0.001**
Cystatin C (mmol/L)	0.86 (0.71–1.04)	0.76 (0.66–0.92)	**<0.001**
Albumin (g/L)	43.00 (41.10–44.60)	43.10 (40.80–45.00)	0.45

Statistically significant values (*p* < 0.05) are written in bold. Legend: BMI—body mass index; SBP—systolic blood pressure; DBP—diastolic blood pressure; HDL-c—high-density cholesterol; LDL-c—low-density cholesterol; TG—triglycerides; HbA1c—glycated hemoglobin; CVI—cardio-vascular insult; TIA—transitory ischemic attack; CABG—coronary artery bypass graft; eGFR—estimated glomerular filtration rate.

**Table 2 biomolecules-14-01584-t002:** Genotype and allele frequencies of the rs2010963 polymorphism.

VEGFA rs2010963	Case (*n* = 484)	Control (*n* = 1105)	*p* Value
CC	49 (10.1%)	90 (8.1%)	0.078
CG	232 (47.9%)	487 (44.1%)	
GG	203 (41.9%)	528 (47.8%)	
C alelle (%) (MAF)	330 (34.1%)	667 (30.2%)	**0.029**
G alelle (%)	638 (65.9%)	1543 (69.8%)	
HWE (*p*-value)	0.14	0.13	

Statistically significant values (*p* < 0.05) are written in bold. Abbreviations: HWE—Hardy–Weinberg equilibrium; MAF—minor allele frequency.

**Table 3 biomolecules-14-01584-t003:** Association between rs2010963 and MI in Slovenian subjects with T2DM.

VEGFA rs2010963	Count	OR (95% CI)	*p* Value for OR
co-dominant			
CC vs. GG	49/90 vs. 203/528	1.48 (0.98–2.22)	0.0612
CG vs. GG	232/487 vs. 203/528	1.29 (1.01–1.64)	**0.0404**
Dominant			
[CC + CG] vs. GG	281/577 vs. 203/528	1.32 (1.05–1.66)	**0.0197**
Recessive			
CC vs. [CG + GG]	49/90 vs. 435/1015	1.3 (0.88–1.91)	0.19
Overdominant			
[CC + GG] vs. CG	252/618 vs. 232/487	0.83 (0.66–1.05)	0.11

Statistically significant values (*p* < 0.05) are written in bold. Adjusted for age, waist circumference, diastolic blood pressure, fasting glucose, total cholesterol, gender, smoking, urea, eGFR, cystatin C, and creatinine.

**Table 4 biomolecules-14-01584-t004:** Serum VEGFA levels in a subpopulation of our participants (*n* = 100).

	Case	Control	*p* Value
VEGFA (pg/mL)	52.46 (44.17–69.95)	50.31 (34.10–76.34)	0.30

**Table 5 biomolecules-14-01584-t005:** Association between rs2010963 genotypes and serum VEGFA levels (*n* = 100).

	CC (MAF)	CG	GG	*p* Value
VEGFA (pg/mL)	57.04 (39.66–59.00)	48.80 (35.09–73.66)	50.65 (37.98–80.90)	0.66

## Data Availability

The data presented in this study are available on request from the corresponding author due to sensitive information (patients’ clinical data).

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
