# Peer review of "The VEGFA rs2010963 Gene Polymorphism Is a Potential Genetic Risk Factor for Myocardial Infarction in Slovenian Subjects with Type 2 Diabetes Mellitus"

_biomolecules, 2024, doi:10.3390/biom14121584_

Round 1
Reviewer 1 Report
Comments and Suggestions for Authors
This is very important study in the field of premature death associated with type 2 diabetes. The study also included a more than sufficient number of patients considering the inclusion criteria.
These are my suggestions to improve paper:
1. Since this study investigate polymorphism of the VEGF gene, it would be interesting to see a difference in albuminuria level, the most sensitive clinical marker of endothelia dysfunction, between of the case and control groups.
2. Why case group subjects with MI had pretty high LDL-cholesterol level (optimal is below 1.4 mmol/L)?
3. In Table 1 please explain what CVI, TIA and CABG abbreviations means.
4. In Methods section authors indicated that the present case-control study included 1589 unrelated Caucasians with T2DM lasting longer than 10 years. However, in Table 1 case group had 11 years of disease duration with interquartile range 6.00 - 20.00 indicating that some patients had shorter duration of disease. Please explain.
5. In table 1 markers of renal function must be indicated. In patients with type 2 diabetes without renal disease 10-year mortality risk is increased for only 4% compared to persons without diabetes or kidney disease. However, in T2DM with eGFR below 60 and albuminuria that risk is 48% (12 times higher). So, when discussing about cardiovascular risk in T2DM, parameters of renal function is essential. In addition, you recently observed that VEGF gene may be associated with diabetic nephropathy in caucasians with type 2 diabetes mellitus.
6. In Introduction, you mentioned AMI without explanation (ref. 3), and after that AMI with explanation (ref. 4). In addition, you must explain all abbreviations like GAWS, PDGF and similar.
7. In Methods section "T2DM was diagnosed according to the criteria of the American Heart Association"!! Please correct.
Author Response
This is very important study in the field of premature death associated with type 2 diabetes. The study also included a more than sufficient number of patients considering the inclusion criteria.
These are my suggestions to improve paper:
- Since this study investigate polymorphism of the VEGF gene, it would be interesting to see a difference in albuminuria level, the most sensitive clinical marker of endothelia dysfunction, between of the case and control groups.
Thank you for this insightful comment. Unfortunately we did not gather information about albuminuria of the participants, nor did we collect urine samples. We included this limitation of our study in the discussion section.
- Why case group subjects with MI had pretty high LDL-cholesterol level (optimal is below 1.4 mmol/L)?
We agree that the lipid status of the participants included in our study was evidently not optimally managed. We hypothesize that adherence to therapy (both pharmacological and non-pharmacological) was the main issue here. We have included a paragraph in the discussion regarding the topic.
- In Methods section authors indicated that the present case-control study included 1589 unrelated Caucasians with T2DM lasting longer than 10 years. However, in Table 1 case group had 11 years of disease duration with interquartile range 6.00 - 20.00 indicating that some patients had shorter duration of disease. Please explain.
Duration of diabetes longer than 10 years was the inclusion criterion only for the participants in the control group. We apologize for not being clear enough.
- In table 1 markers of renal function must be indicated. In patients with type 2 diabetes without renal disease 10-year mortality risk is increased for only 4% compared to persons without diabetes or kidney disease. However, in T2DM with eGFR below 60 and albuminuria that risk is 48% (12 times higher). So, when discussing about cardiovascular risk in T2DM, parameters of renal function is essential. In addition, you recently observed that VEGF gene may be associated with diabetic nephropathy in caucasians with type 2 diabetes mellitus.
We have included several markers of kidney function (eGFR, serum creatinine, urea, cistatin C and albumin) in Table 1 as well as added parts in the discussion regarding the topic. We updated the Table 3 after adjusting for the variables that the groups differed in significantly.
- In Table 1 please explain what CVI, TIA and CABG abbreviations means.
- In Introduction, you mentioned AMI without explanation (ref. 3), and after that AMI with explanation (ref. 4). In addition, you must explain all abbreviations like GAWS, PDGF and similar.
Thank you for noticing this. We have explained all the abbreviations the first time they appear in the text as well as included a Legend explaining the abbreviations used in Table 1.
- In Methods section "T2DM was diagnosed according to the criteria of the American Heart Association"!! Please correct.
We apologise for our carelessness, we have corrected the sentence accordingly: »T2DM was diagnosed according to the criteria of the American Diabetes Association«.
Reviewer 2 Report
Comments and Suggestions for Authors
The present manuscript investigates the association between a VEGF polymorphism with myocardial infarction in a slovenian type 2 diabetic population, as a potential risk factor of a cardiovascular event. The paper is very important in the area, the references are in general up-to-date, and the language seems correct throughout the text.
However, the serum/plasma levels of VEGF-A might have been examinated simultaneously in order to better describe the present clinical model. Finally, i' would like to see a revised form of this manuscript towards this direction.
Author Response
The present manuscript investigates the association between a VEGF polymorphism with myocardial infarction in a slovenian type 2 diabetic population, as a potential risk factor of a cardiovascular event. The paper is very important in the area, the references are in general up-to-date, and the language seems correct throughout the text.
However, the serum/plasma levels of VEGF-A might have been examinated simultaneously in order to better describe the present clinical model. Finally, i' would like to see a revised form of this manuscript towards this direction.
Thank you for the kind and insightful comments. We have determined the serum VEGF levels in a smaller portion of the participants (n=100) and have included the information in the article (Tables 4 and 5) and the text marked in blue.
Round 2
Reviewer 1 Report
Comments and Suggestions for Authors
Thank you for addressing all my concerns.
Reviewer 2 Report
Comments and Suggestions for Authors
The manuscript can be published in its revised form.